# Syphilis seroprevalence and risk factors among first-time blood donors in Brazil: A comprehensive repeated cross-sectional analysis spanning a decade

Andre Lazzeri Cortez[1]*, Vivian I. Avelino-Silva[1,2,3], Barbara Labella Henriques[1], Sebastian Vernal[1], Cesar de Almeida-Neto[4], André Rolim Belisário[5], Paula Loureiro[6], Claudia de Alvarenga Maximo[7], Sheila de Oliveira Garcia Mateos[1,8], Philippe Mayaud[9], Ester Cerdeira Sabino[1]

1 Faculdade de Medicina da Universidade de São Paulo, São Paulo, Brazil, 2 Vitalant Research Institute, San Francisco, California, United States of America, 3 Department of Epidemiology and Biostatistics, University of California San Francisco, San Francisco, California, United States of America, 4 Fundação Pró-Sangue Hemocentro de São Paulo, São Paulo, Brazil, 5 Hemominas, Brazil, 6 Hemope, Brazil, 7 Hemorio, Brazil, 8 Universidade Municipal de São Caetano do Sul, São Caetano do Sul, Brazil, 9 Department of Clinical Research, London School of Hygiene and Tropical Medicine, London, United Kingdom

* andre.cortez@alumni.usp.br

## Abstract

### Background

Syphilis remains a global health challenge, with rising incidence rates worldwide Prevalence surveys conducted in Brazil over extended periods of time are scarce. This study examines the secular trends and risk factors for syphilis seroprevalence among first-time blood donors in Brazil.

### Methods

A retrospective analysis was conducted as part of a multicenter, repeated cross-sectional survey of blood donors from four major Brazilian blood centers, covering the period from 2007 to 2020. First-time donors who had undergone valid treponemal screening tests were included in the final dataset. Demographic characteristics and serological results were analyzed to identify risk factors for syphilis seroprevalence using multivariate Poisson models. An interaction term between age group and donation year was added to the final model. Model comparisons were performed using Likelihood Ratio Tests (LRT) and Akaike Information Criterion (AIC).

### Results

1,424,850 donations from first-time donors were included during the study period. The overall syphilis seroprevalence was 2.19%, with significant heterogeneity across centers. Risk factors for increased seroprevalence included male gender, older age, lower education level, and self-reported black or mixed skin color. Notably, an increasing trend in syphilis

**Data Availability Statement:** "Data are available at https://data.mendeley.com/datasets/5wbg5k4pm2/1".

**Funding:** The author(s) received no specific funding for this work.

**Competing interests:** The authors have declared that no competing interests exist.

seroprevalence was observed among younger donors and those born after 1990. Interaction analyses revealed significant effects between visit period and key demographic variables (age group, gender, education, and ethnicity), with the interaction between age group and donation year indicating higher seroprevalence among younger age groups in recent years.

## Conclusion

The study highlights a high syphilis seroprevalence among first-time blood donors in Brazil, which has significant implications for blood safety and public health. The increasing trend among younger donors suggests a shift towards newer infections, warranting continued surveillance in this demographic.

## Introduction

Syphilis is an infectious disease caused by *Treponema pallidum*, subspecies *pallidum*, a bacterium classified under the *Spirochetes* phylum [1]. Unfortunately, syphilis and its sequelae continue to be prevalent globally, with rising incidence rates [2].

Blood donor screening for syphilis started in the 1940s due to the report of transfusion-transmitted cases [3]. Storage conditions for blood products have further decreased transmission risk [4]. In high-income countries, evidence of sustained, extremely low prevalence among blood donors has pleaded against maintaining universal syphilis screening, particularly because of costs [2]. However, recent syphilis recrudescence in many population subgroups has led blood bank managers to step back and keep universal screening, with a few exceptions [3].

In Brazil, syphilis blood screening is mandatory, and the majority of the blood centers have replaced non-treponemal tests with automated treponemal serological diagnostic methods such as enzyme-linked immunosorbent assays (ELISAs) or chemiluminescence immunoassays (CLIAs).

In 2010, the Brazilian Ministry of Health implemented mandatory notification for acquired syphilis cases. The incidence of acquired syphilis among adolescents (aged 13 to 19) has increased 2.2-fold between 2015 and 2021, according to recent data released in 2022 [4]. Males account for 60.6% of all cases, with those aged 20 to 29 (35.6%) and 30 to 39 (22.3%) being especially vulnerable [4].

Understanding the prevalence trends of syphilis is essential for effective public health interventions. Unfortunately, comprehensive, long-term studies are scarce in Brazil. In this context, leveraging data from blood donors could offer invaluable insights, particularly among individuals with a lower exposure risk for sexually transmitted disease.

Since 2007, the NHLBI REDS program in Brazil has extracted and organized data from four large blood centers with data from more than 4 million donations [5]. We used this dataset to investigate syphilis prevalence trends in Brazil.

## Methods

### Study design, settings and participants

The National Heart, Lung, and Blood Institute Recipient Epidemiology and Donor Evaluation Study (REDS program) included Brazil as an international site in 2007. The Fundação Pró-

Sangue (FPS) in São Paulo (SP), Fundação Hemominas in Belo Horizonte (MG), and Fundação Hemope in Recife (PE) were included at the beginning of the REDS-II phase, while Hemorio in Rio de Janeiro (RJ) was included in 2012 during the REDS-III phase. The current analysis focuses on first-time blood donors from each center, defined as all donors with no previous blood unit identifier record at the time of data extraction. The age range of donors was restricted from 18 to 65 years old at the time of donation, and only results of valid treponemal screening tests were included in this analysis. The data utilized for this research were accessed on September 13, 2022. The study protocol was reviewed and approved by the Comissão Nacional de Ética em Pesquisa (CONEP), the national ethics committee in Brazil. The approval was granted under the permit number 6.865.618, and the date of approval was January 18, 2006. Any amendments to the protocol were also reviewed and approved by the same committee, as documented in the subsequent approvals on June 29, 2019.

## Variables and measurements

The age was defined at the first donation date. Ethnicity was self-reported by the blood donors. Routine screening in Brazil includes serological tests for HIV, hepatitis B (HBV, both core antibodies [HBcAb] and surface antigen [HBsAg]), hepatitis C (HCV), human T-lymphotropic viruses (HTLV)-1 and -2, *Trypanosoma cruzi* (Chagas disease), and *Treponema pallidum* (Syphilis). All centers were using the ARCHITECT Syphilis TP Assay, which is a chemiluminescent microparticle immunoassay (CMIA) for the qualitative detection of antibodies (IgG and IgM) directed against Treponema pallidum. In the context of syphilis, the detection of antibodies through these treponemal methods signifies a history of exposure to the pathogen rather than being indicative of an ongoing active syphilis infection [6]. Treponemal testing was initiated in Rio de Janeiro (RJ) in the year 2012, thereby allowing for the inclusion of data from the year 2013 and subsequent years in the analysis. In contrast, in Pernambuco (PE), the implementation of treponemal testing has been in place since the year 2010.

## Statistical methods

Quantitative variables such as year of donation and age were categorized into discrete groups to help model results interpretation. The donation period was included in the model to analyze secular trends among different sub-groups. To determine syphilis seroprevalence, the total number of syphilis-positive blood donors without any previous donation record was considered the numerator, with the total number of first-time blood donors with valid treponemal results being the denominator. Unadjusted seroprevalence estimates with 95% confidence intervals (CI) were obtained using binomial distribution. We chose to model syphilis seroprevalence ratios according to social demographic variables. We excluded serology results as covariates to avoid the risk of multicollinearity. Multivariate Poisson models were developed to identify individual characteristics that are independently associated with positive results at first-time blood donation. We evaluated the potential interaction between age group and donation year to determine if the effect of age on syphilis seroprevalence varied over different visit years. The primary analysis included an interaction between age group and visit period to assess whether age-specific trends in syphilis seroprevalence varied over time. Model comparisons used Likelihood Ratio Tests (LRT) and Akaike Information Criterion (AIC) to evaluate fit improvements, with significant LRT p-values and lower AIC values indicating a better fit. During peer review, additional post-hoc models were developed to test interactions between visit period and each of the following: declared race, educational level, and gender.

## Results

### Population characteristics

Fig 1 summarizes the study population. During the study period, there were 1,616,746 first-time blood donors; of those, 1,424,850 (88%) had valid serological results for syphilis, with approximately 101,775 first-time blood donors per year (range 45,072–134,348).

Table 1 shows the demographics and serological findings of first-time blood donors by donation center. All blood banks had more male donors, except for Hemominas (MG), which had 50.9% female donors. The majority of donors were between the ages of 18 and 34.

The syphilis seroprevalence was 2.19% (95%CI: 2.17–2.21%), placing it in the top two highest seroreactivity rates seen, similar to the values achieved for HBV anti-HBc biomarkers. Syphilis seroprevalence showed the greatest heterogeneity across different centers, ranging from 1.34% in FPS (Sao Paulo) to 3.50% in Hemope (Pernambuco). HIV seroprevalence was consistently lower across all centers, ranging from 0.18% to 0.41%, with an overall average of 0.2%.

Table 2 provides the observed unadjusted syphilis seroprevalence ratios and 95% confidence intervals (CIs) using binomial distributions for all of the variables indicated in Table 1.

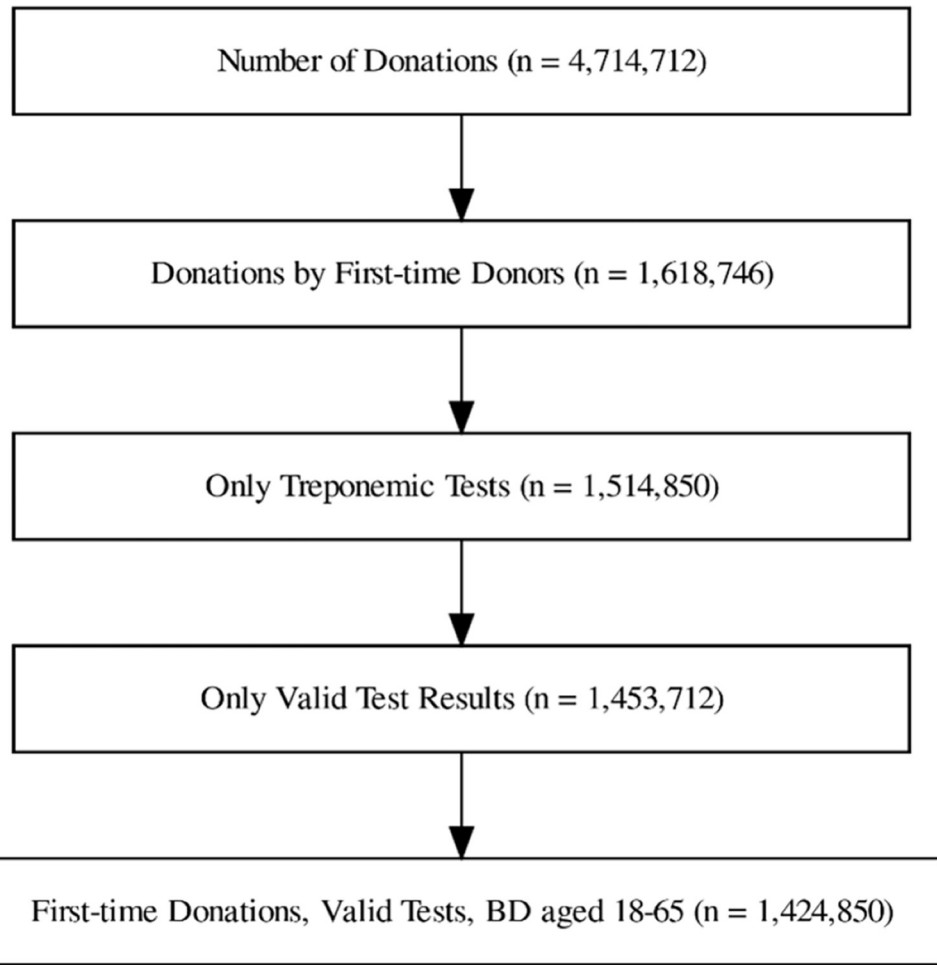

**Fig 1. Flow diagram of participants selection for the analysis of syphilis seroprevalence among first-time blood donors (BD) from the REDS-III Dataset (2007–2020).**

**Table 1. Demographic characteristics and serological results of first-time blood donors in four donation centers in Brazil, REDS-III study (2007–2020).**

| Characteristic | MG | PE | RJ | SP |
|---|---|---|---|---|
| | N = 310,969 | N = 360,899 | N = 210,704 | N = 542,278 |
| | n (%) | n (%) | n (%) | n (%) |
| **Sex** | | | | |
| Female | 158,370 (50.93%) | 139,099 (38.54%) | 100,922 (47.90%) | 267,964 (49.41%) |
| Male | 152,599 (49.07%) | 221,800 (61.46%) | 109,782 (52.10%) | 274,314 (50.59%) |
| **Age Group, year** | | | | |
| 18–24 | 123,153 (39.60%) | 138,282 (38.32%) | 68,369 (32.45%) | 176,728 (32.59%) |
| 25–34 | 111,795 (35.95%) | 119,577 (33.13%) | 67,768 (32.16%) | 193,013 (35.59%) |
| 35–44 | 49,001 (15.76%) | 66,666 (18.47%) | 41,933 (19.90%) | 101,481 (18.71%) |
| 45–54 | 21,633 (6.96%) | 29,469 (8.17%) | 23,465 (11.14%) | 53,105 (9.79%) |
| 55–65 | 5,387 (1.73%) | 6,905 (1.91%) | 9,169 (4.35%) | 17,951 (3.31%) |
| **Ethnicity*** | | | | |
| Black | 47,392 (15.24%) | 33,901 (9.39%) | 18,546 (8.80%) | 40,266 (7.43%) |
| Mixed | 156,416 (50.30%) | 191,817 (53.15%) | 98,880 (46.93%) | 168,949 (31.16%) |
| White | 100,254 (32.24%) | 84,302 (23.36%) | 92,044 (43.68%) | 291,392 (53.73%) |
| Other | 2,710 (0.87%) | 1,379 (0.38%) | 866 (0.41%) | 11,008 (2.03%) |
| **Educational Level*** | | | | |
| No Formal Education | 19,565 (6.29%) | 2,640 (0.73%) | 225 (0.11%) | 1,350 (0.25%) |
| Primary | 56,527 (18.18%) | 70,299 (19.48%) | 18,009 (8.55%) | 69,763 (12.86%) |
| Secondary | 171,802 (55.25%) | 191,662 (53.11%) | 97,075 (46.07%) | 297,123 (54.79%) |
| University/Post- Graduate | 60,605 (19.49%) | 46,846 (12.98%) | 95,395 (45.27%) | 143,605 (26.48%) |
| **Donation Year** | | | | |
| 2007–2009 | 67,746 (21.79%) | - | - | 104,686 (19.30%) |
| 010–2012 | 70,178 (22.57%) | 97,513 (27.02%) | - | 125,220 (23.09%) |
| 2013–2016 | 94,076 (30.25%) | 141,732 (39.27%) | 73,411 (34.84%) | 159,740 (29.46%) |
| 2017–2020 | 78,969 (25.39%) | 121,654 (33.71%) | 137,293 (65.16%) | 152,632 (28.15%) |
| **Positive Serologic Biomarkers** | | | | |
| Syphilis | 4,884 (1.57%) | 12,615 (3.50%) | 6,443 (3.06%) | 7,263 (1.34%) |
| HIV | 546 (0.18%) | 719 (0.20%) | 869 (0.41%) | 677 (0.12%) |
| HCV | 634 (0.20%) | 1,436 (0.40%) | 852 (0.40%) | 2,210 (0.41%) |
| HBcAb | 6,626 (2.13%) | 12,022 (3.33%) | 4,974 (2.36%) | 10,802 (1.99%) |
| HBsAg | 617 (0.20%) | 1,040 (0.29%) | 709 (0.34%) | 897 (0.17%) |
| *T. cruzi* | 350 (0.11%) | 356 (0.10%) | 1,150 (0.55%) | 911 (0.17%) |
| HTLV | 530 (0.17%) | 1,111 (0.31%) | 715 (0.34%) | 857 (0.16%) |

*Ethnicity and Educational Level for 30,663 (5.65%) and 30,437 (5.61%) participants, respectively

MG = Hemominas; PE = Hemope; RJ = Hemorio; SP = Sao Paulo Fundação Pró-Sangue (FPS)

Male donors had higher prevalences in all four donation centers. Additionally, treponemal seropositivity increased with age and was higher among donors with lower education levels and those who self-reported as black or mixed. Syphilis seroprevalence was also much higher among blood donors who tested positive for other infectious diseases; for example, syphilis rates among HIV-positive individuals ranged from 9.21% (95CI: 7.28–11.13) in Rio de Janeiro to 16.13% (95% CI: 13.44–18.82) in Recife/Pernambuco.

Syphilis seroprevalence trends over the study period are described according to age group (Fig 2) and decade of birth (Fig 3). The prevalence is increasing in younger individuals and those born after 1990.

**Table 2. Estimates of syphilis seroprevalence in first-time blood donors, according to selected demographic variables, donation center and other serological results in Brazil, REDS-III study (2007–2020).**

| Characteristic | MG | PE | RJ | SP |
|---|---|---|---|---|
| | % (95%CI) | % (95%CI) | % (95%CI) | % (95%CI) |
| **Sex** | | | | |
| Female | 1.25 (1.19–1.30) | 2.81 (2.72–2.90) | 2.77 (2.67–2.88) | 1.11 (1.07–1.15) |
| Male | 1.90 (1.84–1.97) | 3.93 (3.85–4.01) | 3.32 (3.22–3.43) | 1.56 (1.52–1.61) |
| **Age Group, years** | | | | |
| 18–24 | 0.73 (0.69–0.78) | 1.48 (1.42–1.54) | 1.89 (1.78–1.99) | 0.69 (0.66–0.73) |
| 25–34 | 1.19 (1.13–1.26) | 2.86 (2.76–2.95) | 2.97 (2.85–3.10) | 1.01 (0.97–1.06) |
| 35–44 | 2.24 (2.12–2.38) | 5.01 (4.85–5.18) | 3.43 (3.26–3.61) | 1.55 (1.48–1.63) |
| 45–54 | 5.06 (4.77–5.36) | 9.56 (9.23–9.90) | 4.56 (4.30–4.83) | 3.01 (2.87–3.16) |
| 55–65 | 8.41 (7.68–9.18) | 14.42 (13.60–15.28) | 6.88 (6.37–7.42) | 5.02 (4.71–5.35) |
| **Ethnicity** | | | | |
| Black | 1.87 (1.75–2.00) | 4.77 (4.55–5.01) | 5.34 (5.02–5.68) | 1.71 (1.59–1.85) |
| Mixed | 1.58 (1.52–1.65) | 3.60 (3.52–3.68) | 3.46 (3.35–3.58) | 1.58 (1.52–1.64) |
| White | 1.40 (1.33–1.48) | 2.45 (2.34–2.55) | 2.14 (2.05–2.23) | 1.10 (1.06–1.14) |
| Other | 1.29 (0.90–1.79) | 3.19 (2.33–4.26) | 2.89 (1.88–4.23) | 0.68 (0.54–0.85) |
| **Educational Level** | | | | |
| No Formal Education | 3.21 (2.97–3.47) | 10.61 (9.46–11.84) | 10.67 (6.95–15.45) | 5.33 (4.20–6.67) |
| Primary | 2.91 (2.77–3.05) | 6.48 (6.30–6.66) | 7.17 (6.80–7.56) | 2.84 (2.72–2.97) |
| Secondary | 1.17 (1.12–1.23) | 2.69 (2.62–2.77) | 3.61 (3.50–3.73) | 1.14 (1.10–1.18) |
| University/Post- Graduate | 0.89 (0.82–0.97) | 1.34 (1.24–1.45) | 1.70 (1.62–1.78) | 0.83 (0.78–0.87) |
| **Donation Year** | | | | |
| 2007–2009 | 2.00 (1.89–2.10) | - | - | 1.23 (1.17–1.30) |
| 2010–2012 | 1.47 (1.38–1.56) | 4.16 (4.03–4.28) | - | 1.39 (1.33–1.46) |
| 2013–2016 | 1.30 (1.23–1.37) | 3.47 (3.37–3.57) | 3.03 (2.91–3.16) | 1.37 (1.31–1.42) |
| 2017–2020 | 1.62 (1.54–1.71) | 3.00 (2.90–3.09) | 3.07 (2.98–3.16) | 1.34 (1.28–1.40) |
| **Positive Serologic Biomarkers** | | | | |
| HIV | 11.90 (9.31–14.92) | 16.13 (13.52–19.03) | 9.21 (7.37–11.33) | 10.93 (8.68–13.53) |
| HCV | 4.73 (3.22–6.69) | 8.29 (6.91–9.83) | 5.28 (3.88–7.00) | 3.35 (2.64–4.19) |
| HbcAb | 7.15 (6.54–7.80) | 12.69 (12.10–13.29) | 9.83 (9.02–10.69) | 6.11 (5.67–6.58) |
| HBsAg | 4.21 (2.77–6.11) | 8.37 (6.75–10.22) | 4.23 (2.87–5.99) | 4.01 (2.83–5.51) |
| *T. cruzi* | 6.00 (3.75–9.03) | 5.34 (3.24–8.21) | 2.35 (1.55–3.40) | 4.06 (2.88–5.55) |
| HTLV | 4.15 (2.62–6.22) | 8.46 (6.89–10.25) | 7.55 (5.72–9.74) | 4.32 (3.06–5.90) |

MG = Hemominas; PE = Hemope; RJ = Hemorio; SP = Sao Paulo Fundação Pró-Sangue (FPS)

Results from the multivariate Poisson regression model that included an interaction term for age group and donation year are displayed in Table 3. Fig 4 illustrates model predictions of syphilis seroprevalences, highlighting the interaction effect found between the demographic variables and period of donation. Model comparisons using Likelihood Ratio Tests (LRT) and Akaike Information Criterion (AIC) indicated that adding individual interaction terms to the base model (without interactions) significantly improved fit. The age group x visit period interaction showed a statistically significant improvement (ΔDeviance = 88.3, p < 0.001). Other tested interactions were also significant, including visit period x declared race (ΔDeviance = 88.3, p < 0.001), visit period x educational level (ΔDeviance = 207.0, p < 0.001), and visit period x gender (ΔDeviance = 38.1, p < 0.001), each showing reduced AIC values.

The prevalence of syphilis continued to be associated with factors such as skin color, poor level of education, and male gender. The decrease in overall prevalence in recent years appears

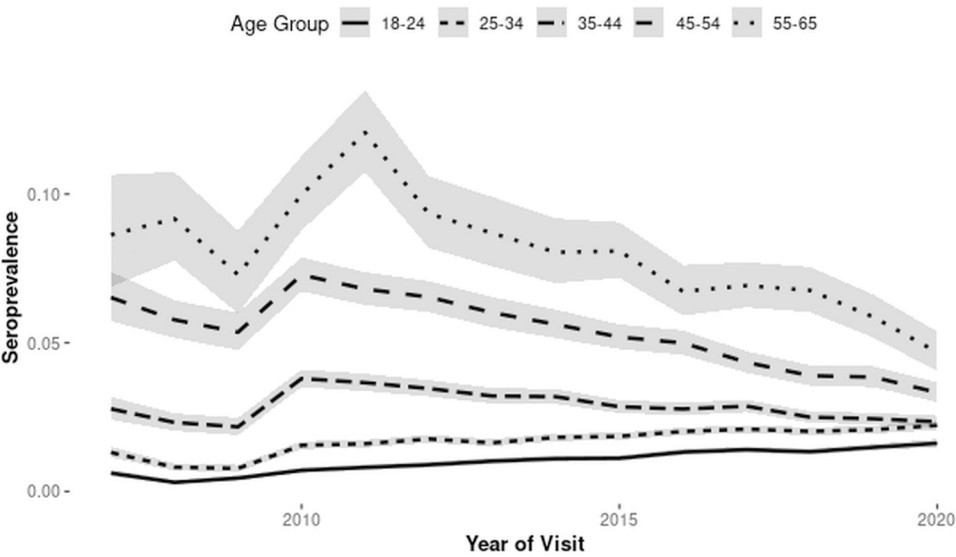

**Fig 2. Unadjusted syphilis seroprevalences with 95% CI (grey shading) according to age group and donation year.**

to be largely driven by a decline in syphilis seroprevalence among older age groups, particularly those aged 55–65. In contrast, the model indicates a slight increasing trend in syphilis seroprevalence among younger donors, specifically those aged 18–34, as depicted in Fig 4.

## Discussion

In this retrospective, pluriannual cross-sectional analysis, we explored the seroprevalence and risk factors of syphilis among first-time blood donors in four centers in Brazil over the period 2007–2020. Overall, the seroprevalence of syphilis was 2.19% (95% CI: 2.17–2.21%), the highest among all serologic markers and the leading cause of serological deferral.

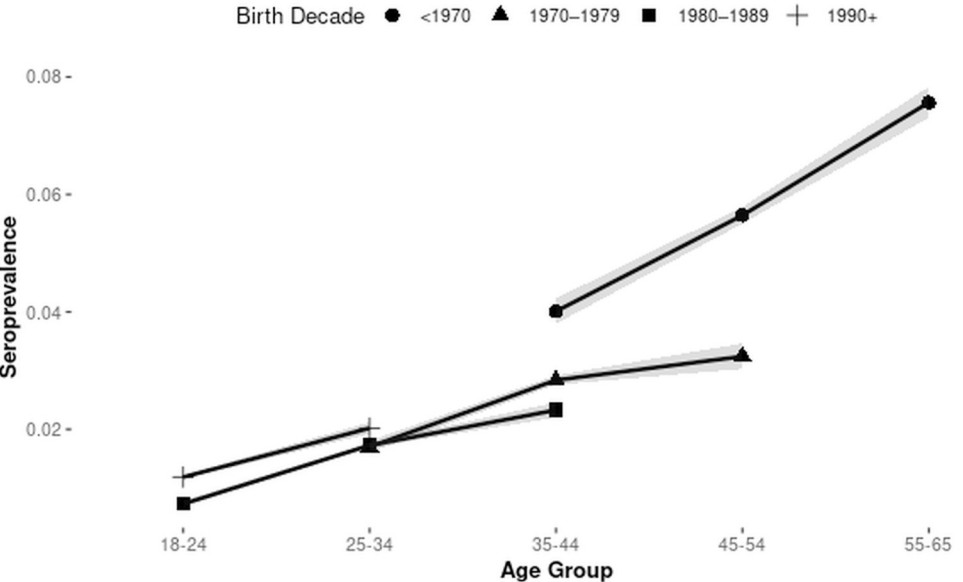

**Fig 3. Unadjusted syphilis seroprevalences with 95% CI (grey shading) according to birth decade and age group.**

**Table 3. Adjusted syphilis seroprevalence ratios among first-time blood donors in Brazil, REDS-III study (2007–2020), results from multivariable Poisson model with interaction term between age group and donation year.**

| Characteristic | Seroprevalence Ratio (95%CI) | p-value |
|---|---|---|
| **Sex** | | |
| Female | 1.0 | |
| Male | 1.26 (1.23–1.29) | <0.001 |
| **Age Group, years** | | |
| 18–25 | 1.0 | |
| 25–34 | 2.09 (1.82–2.41) | <0.001 |
| 35–44 | 5.14 (4.46–5.90) | <0.001 |
| 45–54 | 12.72 (11.14–14.53) | <0.001 |
| 55–65 | 18.26 (15.58–21.39) | <0.001 |
| **Self reported ethnicity** | | |
| White | 1.0 | |
| Mixed | 1.31 (1.28–1.35) | <0.001 |
| Black | 1.58 (1.52–1.64) | <0.001 |
| Other | 0.96 (0.83–1.12) | 0.63 |
| **Educational Level** | | |
| No formal education | 4.03 (3.74–4.35) | <0.001 |
| Primary | 2.81 (2.70–2.93) | <0.001 |
| Secondary | 1.82 (1.76–1.89) | <0.001 |
| University/Post Graduate | 1.0 | |
| **Donation Center** | | |
| MG (Hemominas) | 1.0 | |
| PE (Hemope) | 1.93 (1.86–1.99) | <0.001 |
| RJ (Hemorio) | 2.17 (2.08–2.26) | <0.001 |
| SP (Fundação Pró-Sangue (FPS)) | 0.89 (0.86–0.92) | <0.001 |
| **Donation period** | | |
| 2007–2009 | 1.0 | |
| 2010–2012 | 0.96 (0.84–1.09) | 0.50 |
| 2013–2016 | 0.68 (0.60–0.77) | <0.001 |
| 2017–2020 | 0.54 (0.48–0.61) | <0.001 |

The prevalence of treponemal seropositivity rates in blood donors is higher compared to other studies conducted in Brazil. These studies reported rates of 0.87% in Goiânia [5], 1.09% in southwest Bahia [7], and 0.14% in Santa Catarina [8]. Research conducted in high-income countries has indicated even lower rates of occurrence, such as 0.16% in the United States [9], 0.03% in Italy [10], and 0.4% in Qatar [11].

A possible explanation for the higher prevalence in our analysis is that we only considered first-time donors, thus reducing the selection bias and dilution effect of including repeat donors, as only seronegative blood donors would be invited to donate again. Other Brazilian studies have used non-treponemal assays for syphilis measurement [12], which would under-estimate treponemal exposure.

In the multivariate Poisson regression analysis, the seroprevalence of treponemal infection increased with age, with the greatest prevalence ratios seen in those older than 56 years old, as would be predicted for a biomarker that assesses lifelong exposure [13]. The results are consistent with other Brazilian studies [14]. However, when factoring in an interaction effect between age group and donation year period, we found that younger first-time blood donors (Ages 18–24 and 25–34) had actually an increasing treponemal seroprevalence trend (Fig 4).

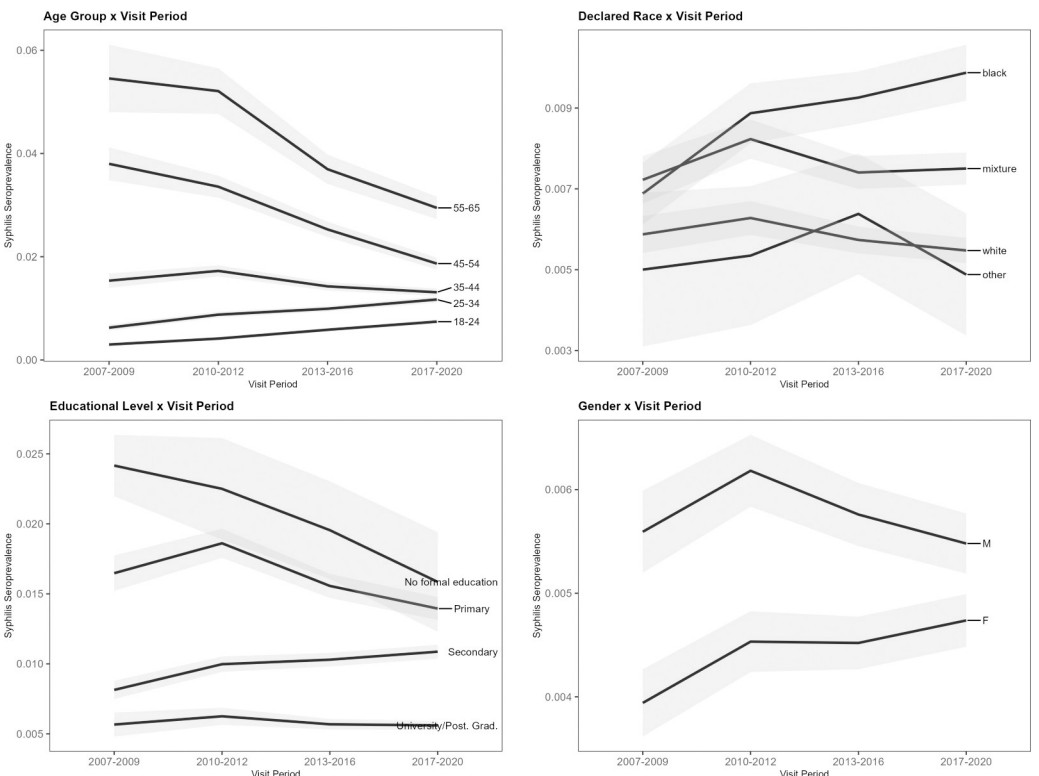

**Fig 4. Estimated marginal means of predicted syphilis seroprevalence over time by age group with 95% confidence intervals (gray shading).**

This observation is consistent with data from the broader Brazilian population, where syphilis incidence rates have escalated from 9.3 per 100,000 inhabitants in 2011 to 44.6 per 100,000 in 2017 [4]. Notably, during the period from 2015 to 2021, the incidence of reported acquired syphilis cases among Brazilian adolescents has risen 2.2-fold [4].

Further, we conducted post-hoc analyses to evaluate additional interactions between visit period and other sociodemographic factors during peer review. Each interaction term, including visit period with declared race, education, and gender, significantly improved model fit. These findings highlight distinct syphilis prevalence trends among different demographic subgroups, suggesting that specific populations may be disproportionately affected by syphilis over time.

Male donors had significantly higher syphilis seroprevalence than female donors (PR 1.26, 95% CI 1.23–1.29). Other studies from Brazil in the last decade have also shown higher syphilis rates among men [15, 16]. A systematic review and meta-analysis evaluating 23 studies from Ethiopia also reported that male blood donors were more likely to be seroreactive for syphilis compared to female blood donors (OR 1.35, 95%CI: 1.01–1.79) [17]. National data from the Brazilian Ministry of Health further indicate that while the male-to-female syphilis ratio was approximately 17 men for every 10 women in 2021, this pattern reverses among adolescents, with a higher prevalence observed in young females (7 men for every 10 women) [4]. Although male blood donors in our sample had a higher seroprevalence overall, the observed increase in syphilis seroprevalence among younger female donors aligns with this national trend in adolescents, underscoring the rising burden of syphilis among young women in Brazil.

Education level was inversely associated with treponemal seropositivity, with blood donors with primary level education having a much higher risk of infection (PR 2.81, 95%CI: 2.70–

2.927) compared to the donors with university or post-graduate degrees, in agreement with other Brazilian studies [15]. Similar findings were reported from a single blood center in China from 2010 to 2019 [18] and in Ethiopia in 2021 [17]. Low educational levels may be related to lower awareness about disease exposure, prevention, and treatment and scarcer financial resources to access testing and treatment [19].

Of note, black and mixed populations had a significantly higher prevalence than white, even after controlling for education level, confirming the notion that syphilis is a marker of ethnoracial inequalities in Brazil [20]. Our post-hoc analysis also highlighted a rising trend in syphilis seroprevalence among Black self reported ethnicity donors over the study period.

Our study had a number of limitations. First, the treponemal-positive samples were not submitted to additional confirmatory tests, in particular, to determine active or recent syphilis, as these are no longer routinely done in large blood donation centers. Although we observed an increasing trend in seroprevalence among younger donors, the lack of longitudinal cohort data limits our ability to accurately estimate incidence rates. Further analysis including repeating donors, with the use of a person-year denominator, would allow for the calculation of incidence rates, providing a more precise assessment of recent syphilis transmission dynamics. Second, we acquired the data from four metropolitan urban areas situated in state capitals, which makes the information less generalizable to entire states or other regions of Brazil. However, our study had notable strengths, including a large sample size, and the use of interaction analyses to identify nuanced associations between syphilis seroprevalence and demographic factors over time. By aligning our results with national data, our sample appears to represent a reliable sentinel surveillance group for what might happen epidemiologically in the general population [21].

In conclusion, this study found that treponemal seroprevalence was higher among first-time blood donors who were older, male, non-white, and had a lower educational level. Additionally, we observed an increasing trend over the last decade in treponemal seroprevalence among younger blood donors, which might be more closely connected to recent acquisitions, as suggested by the contemporary increase in nationally reported data. This study thus supports the use of treponemal testing results from first-time blood donors to complement surveillance data on the epidemiology of syphilis in the general population.

## Acknowledgments

Special acknowledgment goes to Vivian I. Avelino-Silva, whose guidance in study design was invaluable, ensuring that our research methodology was robust and comprehensive. The expertise brought by Brian Custer alongside Vivian has significantly shaped the framework of our investigation.

Andre Lazzeri Cortez, the main author of this study, performed the crucial statistical analysis, laying the foundation for our interpretations and conclusions. The writing process benefited greatly from the efforts of Barbara Labella Henriques and Sebastian Andres Vernal Carranza. Their ability to articulate complex scientific concepts into comprehensible text has greatly enhanced the quality of our manuscript.

The final review of our study was enriched by the senior expert opinions of Ester Sabino and Philippe Mayaud, assuming a vital role in the review process.

We owe a debt of gratitude to Cesar de Almeida-Neto, MD, PhD, Paula Loureiro, MD, PhD, André Rolim Belisário, PhD, and Claudia de Alvarenga Maximo, MD, for their diligent work in data collection across different blood banks. Their dedication to gathering accurate and comprehensive data has been essential to our study's integrity.

Sheila de Oliveira Garcia Mateos, PhD, as the REDS project coordinator in Brazil, played a dual role in this research. Her coordination efforts ensured smooth operation and data

collection, and her contributions to reviewing the article's findings have been invaluable in refining our conclusions.

## Author Contributions

**Conceptualization:** Vivian I. Avelino-Silva, Cesar de Almeida-Neto, André Rolim Belisário, Claudia de Alvarenga Maximo, Sheila de Oliveira Garcia Mateos, Ester Cerdeira Sabino.

**Data curation:** Vivian I. Avelino-Silva, Cesar de Almeida-Neto, André Rolim Belisário, Claudia de Alvarenga Maximo, Sheila de Oliveira Garcia Mateos, Ester Cerdeira Sabino.

**Formal analysis:** Andre Lazzeri Cortez, Vivian I. Avelino-Silva, Barbara Labella Henriques, Philippe Mayaud.

**Funding acquisition:** Sheila de Oliveira Garcia Mateos, Ester Cerdeira Sabino.

**Investigation:** Andre Lazzeri Cortez, Sheila de Oliveira Garcia Mateos, Ester Cerdeira Sabino.

**Methodology:** Andre Lazzeri Cortez, Vivian I. Avelino-Silva, Sebastian Vernal, Sheila de Oliveira Garcia Mateos, Philippe Mayaud, Ester Cerdeira Sabino.

**Project administration:** Vivian I. Avelino-Silva, Sheila de Oliveira Garcia Mateos, Ester Cerdeira Sabino.

**Resources:** Cesar de Almeida-Neto, André Rolim Belisário, Paula Loureiro, Claudia de Alvarenga Maximo, Ester Cerdeira Sabino.

**Supervision:** Sheila de Oliveira Garcia Mateos, Philippe Mayaud, Ester Cerdeira Sabino.

**Validation:** Cesar de Almeida-Neto, André Rolim Belisário, Claudia de Alvarenga Maximo, Sheila de Oliveira Garcia Mateos, Ester Cerdeira Sabino.

**Visualization:** Sebastian Vernal, Sheila de Oliveira Garcia Mateos, Ester Cerdeira Sabino.

**Writing – original draft:** Andre Lazzeri Cortez, Vivian I. Avelino-Silva, Barbara Labella Henriques, Sebastian Vernal, Sheila de Oliveira Garcia Mateos, Philippe Mayaud, Ester Cerdeira Sabino.

**Writing – review & editing:** Vivian I. Avelino-Silva, Barbara Labella Henriques, Sebastian Vernal, André Rolim Belisário, Paula Loureiro, Sheila de Oliveira Garcia Mateos, Philippe Mayaud, Ester Cerdeira Sabino.

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
