## [Decision Letter · Decision Letter 0]

2 Sep 2024

PONE-D-24-14299Syphilis Seroprevalence and Risk Factors Among First-Time Blood Donors in Brazil:  A Comprehensive Repeated Cross-Sectional Analysis spanning a decadePLOS ONE

Dear Dr. CORTEZ,

Thank you for submitting your manuscript to PLOS ONE. After careful consideration, we feel that it has merit but does not fully meet PLOS ONE’s publication criteria as it currently stands. Therefore, we invite you to submit a revised version of the manuscript that addresses the points raised during the review process.

Both reviewers acknowledged the relevance and soundness of the reported study, making suggestions with the aim of further improving it. Please, attempt to amply incorporate the suggestions made.

We look forward to receiving your revised manuscript.

Kind regards,

Albert Schriefer, M.D., Ph.D.

Section Editor

PLOS ONE

Reviewers' comments:

Reviewer's Responses to Questions

**Comments to the Author**

1. Is the manuscript technically sound, and do the data support the conclusions?

Reviewer #1: Yes

Reviewer #2: Yes

2. Has the statistical analysis been performed appropriately and rigorously? 

Reviewer #1: Yes

Reviewer #2: Yes

3. Have the authors made all data underlying the findings in their manuscript fully available?

Reviewer #1: No

Reviewer #2: Yes

4. Is the manuscript presented in an intelligible fashion and written in standard English?

Reviewer #1: Yes

Reviewer #2: Yes

5. Review Comments to the Author

Reviewer #1: Introduction

Add the reference [4] for “The incidence of acquired syphilis among adolescents (aged 13 to 19) has increased 2.2-fold between 2015 and 2021, according to recent data released in 2022” and remove it from the next sentence.

In the sentence “Routine screening in Brazil includes serological tests for HIV, hepatitis B (HBV, both core antibodies [HBcAb] and surface antigen [HBsAg]), hepatitis C (HCV), human T-lymphotropic viruses (HTLV)-1 and -2, Trypanosoma cruzi (Chagas disease), and syphilis”, add “T. pallidum” before “syphilis” and put the latter within brackets, as you did for T. cruzi, to maintain consistency of listing the pathogens first and the diseases they cause second.

Add “being” after “than” in “In the context of syphilis, the detection of antibodies through these treponemal methods signifies a history of exposure to the pathogen rather than indicative of an ongoing active syphilis infection [6]” or rephrase in another way.

Statistical methods

You say “To determine syphilis seroprevalence, the total number of syphilis-positive blood donors without any previous donation record was considered the numerator, with the total number of first-time blood donors with valid treponemal results being the denominator.” Did you consider calculating syphilis incidence for first-time donors by extrapolating from the repeat donor incidence – a method already used in transfusion risk research? The trouble with syphilis seroprevalence is that it cannot inform how recently the infection occurred. In contrast, incidence estimation with a person-year denominator goes directly at the heart of your research question on secular trends. Although you can infer syphilis incidence from sequential prevalence estimates, it is an indirect and less precise method as it lacks the infection recency parameter. Older donors had longer exposure to T. Pallidum, but you cannot tell at what age they acquired the infection from the prevalence data. For younger donors, this uncertainty is less acute, but it may still be years or even more than a decade apart between the infection and blood donation. This is something to be addressed in the discussion.

“We excluded serology results as covariates to avoid the risk of multicollinearity” sounds odd. Why would rare co-infections, typically with HBV and/or HIV, induce collinearity?

You say “Subsequently, model comparisons were conducted using an Analysis of Variance (ANOVA) to test the significance of the added interaction term, with a p-value threshold of 0.05”, but it seems at odds with Poisson regression mentioned earlier. Was it a post-hoc regression analysis or was the term included in the regression model? Can you write down the equation? Which software did you use?

Results

Define the algorithm for “valid serological results for syphilis”. The donors with either treponemic or non-treponemic test results were discarded from the statistical analysis; if so, how different were their sociodemographic characteristics and serologic testing results from those included in the study?

In Table 2, the phrase “Estimates of Syphilis Seroprevalence Rates” should drop the word “Rates” because it implies a person-time denominator, which was not used here.

In Table 3, “ref.” should be substituted by “1.00”. Although the value is implied in the seroprevalence ratio description, it is better to state it explicitly than to keep some readers guessing. 

“Predicted Syphilis Seroprevalences” in Figure 4 are Poisson model-based marginal means or proportions?

It is unclear to me what was meant by “According to the model, despite Figure 4, there is an increase in prevalence among persons aged 18–34”. Why “despite Figure 4”? Also, you say “The decrease in overall prevalence in recent years is to be attributed to the decline in older strata”, and I suppose this means that the older age groups reduced their prevalence over time as opposed to the younger blood donors. Please clarify and rephrase.

Discussion

In the reference [8], the overall syphilis seroprevalence was 0.14%, not 0.13% as reported in the present study. More importantly, the prevalence was 0.19% among the first-time donors, so this is the value directly comparable with the present study prevalence. This is important in light of your statement “A possible explanation for the higher prevalence in our analysis is that we only considered first-time donors, thus reducing the selection bias and dilution effect of including repeat donors, as only seronegative blood donors would be invited to donate again.” Other southern Brazilian states (e.g. Paraná) also reported lower syphilis seroprevalence among first-time blood donors [8], so systematic regional variations may be a more plausible explanation.

Also, the two-step blood screening that starts with a treponemal test, followed by a non-treponemal test, hugely reduced false positive donor deferral in Brazil, thus reducing the “dilution effect of including repeat donors” (Baião A. et al. in Transfusion Medicine 2014; 24(1):64-66, doi: 10.1111/tme.12095). Resuming, a more thorough discussion of alternative explanations is due.

Age-by-sex secular trends, adjusted for other covariates in multivariable Poisson regression, should be added to better understand how and for whom these trends are changing over time. This is equivalent to adding sex to the interaction term or just a post-hoc regression analysis of marginal means for the age-by-sex groups.

The inclusion of only first-time blood donors in statistical analysis was considered a “notable strength” by the authors. However, it has many drawbacks in comparison with incidence estimation, already mentioned in my comments on statistical methods. Although the incidence estimation methods for first-time donors have their caveats, they rest on solid statistical principles, like so-called raking. I could not find sufficient arguments for favoring serial prevalence over incidence in the present study, nor did I see why first-time donors would be “a more reliable sentinel surveillance group for what might happen epidemiologically in the general population [21].” Behavioral factors are hugely important in motivating people to donate blood and hold the key to the understanding of donor candidate (auto)selection. Resuming, I suggest a better elaboration of the arguments stated for the cited recommendation.

Reviewer #2: The study was conducted with rigor, utilizing a repeated cross-sectional and multicenter design, which allowed for the collection of data from a large sample of blood donors over a significant period (2007-2020). The inclusion of over 1.4 million first-time donors strengthens the robustness of the data. The use of multivariate Poisson models, including interaction terms, demonstrates an advanced and appropriate statistical approach to analyze syphilis seroprevalence and identify associated risk factors. The comparison of models using ANOVA reinforces the validity of the analyses. The conclusions are drawn from the presented data, such as the increase in syphilis prevalence among younger donors, particularly those born after 1990, indicating a concerning trend that requires continuous surveillance. Overall, the manuscript meets the criteria for technically sound scientific research, with data that adequately support the conclusions.

Yes, the statistical analysis was performed appropriately and rigorously. The study utilized multivariate Poisson models, which are suitable for analyzing syphilis seroprevalence and identifying associated risk factors. The inclusion of interaction terms, such as between age group and donation year, adds depth to the analysis and allows for a more nuanced interpretation of trends over time. Furthermore, the model comparisons using ANOVA to evaluate the significance of these interaction effects demonstrate a rigorous approach. The improvement in the adjusted R² from 0.10 to 0.11 and the associated p-value of less than 0.001 reflect the robustness of the statistical analysis.

6. PLOS authors have the option to publish the peer review history of their article (what does this mean?). If published, this will include your full peer review and any attached files.

Reviewer #1: **Yes: **Emil Kupek

Reviewer #2: No

---

## [Author Response · Author response to Decision Letter 0]

17 Nov 2024

I really want to give special thanks to the reviewers that helped us to find interesting new insights about our data.

---

## [Decision Letter · Decision Letter 1]

4 Dec 2024

Syphilis Seroprevalence and Risk Factors Among First-Time Blood Donors in Brazil:  A Comprehensive Repeated Cross-Sectional Analysis spanning a decade

PONE-D-24-14299R1

Dear Dr. CORTEZ,

We’re pleased to inform you that your manuscript has been judged scientifically suitable for publication and will be formally accepted for publication once it meets all outstanding technical requirements.

Kind regards,

Albert Schriefer, M.D., Ph.D.

Section Editor

PLOS ONE

Additional Editor Comments (optional):

Reviewers' comments:

Reviewer's Responses to Questions

**Comments to the Author**

1. If the authors have adequately addressed your comments raised in a previous round of review and you feel that this manuscript is now acceptable for publication, you may indicate that here to bypass the “Comments to the Author” section, enter your conflict of interest statement in the “Confidential to Editor” section, and submit your "Accept" recommendation.

Reviewer #1: All comments have been addressed

Reviewer #2: All comments have been addressed

2. Is the manuscript technically sound, and do the data support the conclusions?

Reviewer #1: Yes

Reviewer #2: Yes

3. Has the statistical analysis been performed appropriately and rigorously? 

Reviewer #1: Yes

Reviewer #2: Yes

4. Have the authors made all data underlying the findings in their manuscript fully available?

Reviewer #1: No

Reviewer #2: Yes

5. Is the manuscript presented in an intelligible fashion and written in standard English?

Reviewer #1: Yes

Reviewer #2: Yes

6. Review Comments to the Author

Reviewer #1: The manuscript is now greatly improved and I am satisfied that the necessary modifications have been made. I have two additional suggestions that I consider nonessential, but could improve a reader’s understanding of the paper.

In the penultimate sentence of the methods, you say “…the detection of antibodies … signifies a history of exposure to the pathogen rather than being indicative of an ongoing active syphilis infection." However, in Table 1 you presented about ten times higher syphilis compared with T. cruzi prevalence, without specifying whether you meant infection or disease. The latter starts with infection but CLIA-positive test results alone cannot distinguish between active infection, past infection, or symptomatic disease. Please clarify under Table 1 what is the difference between the “Syphilis” and T. cruzi rows. Which of the two represents a CMIA-positive test result?

On Figure 2, vertical axis should add “%” to define the scale. Also, “year of visit” was in fact the year of blood donation. There are many other reasons to visit a blood bank but it is the donation that really matters here. It would be a better phrase to use throughout the text, too.

Reviewer #2: The manuscript is clear, well-structured, and scientifically grounded. It presents a balanced discussion, highlighting both the strengths and limitations of the study. The recommendations based on the findings reinforce the practical utility of the research in enhancing epidemiological surveillance and planning public health interventions. I recommend the publication of the manuscript, considering its high relevance, methodological rigor, and significant contribution to understanding the epidemiology of syphilis in Brazil. This is a high-quality study that can positively impact the formulation of health policies and screening programs at a national level.

7. PLOS authors have the option to publish the peer review history of their article (what does this mean?). If published, this will include your full peer review and any attached files.

Reviewer #1: **Yes: **Emil Kupek

Reviewer #2: No

---

## [Editor Report · Acceptance letter]

10 Dec 2024

PONE-D-24-14299R1 

PLOS ONE

Dear Dr. CORTEZ, 

I'm pleased to inform you that your manuscript has been deemed suitable for publication in PLOS ONE. Congratulations! Your manuscript is now being handed over to our production team.

Kind regards, 

on behalf of

Dr. Albert Schriefer 

Section Editor

PLOS ONE